# Identification of potentially effective drugs for metabolic dysfunction-associated steatotic liver disease against liver cirrhosis: In-silico drug repositioning-based retrospective cohort study

Chae Won Lee[1☺], Eun Seok Kang[1,2☺], Seogsong Jeong[2*], Hyun Wook Han[1*]

1 Department of Biomedical Informatics, CHA University School of Medicine, Seongnam, Republic of Korea, 2 Department of Biomedical Informatics, Korea University College of Medicine, Seoul, Republic of Korea

☺ These authors contributed equally to this work.
* stepano7@gmail.com (H.W.H); seogsongjeong@korea.ac.kr (S.J)

## Abstract

### Background

Metabolic dysfunction-associated steatotic liver disease (MASLD) is a major risk factor for liver cirrhosis, yet effective prevention or treatment strategies remain limited. To address this, we utilized a signature-based in silico drug repositioning approach to identify potential therapeutics for MASLD that may reduce the risk of cirrhosis.

### Methods

We analyzed gene expression datasets to identify differentially expressed genes (DEGs) in MASLD and matched them to candidate drugs using L1000CDS2. We further validated potential drugs by cross-referencing with prescription data from the Korea National Health Insurance Service (NHIS). Participants who underwent health screenings between 2013 and 2014 were included. MASLD was diagnosed in individuals with hepatic steatosis (fatty liver index ≥60) and at least one cardiometabolic risk factor.

### Results

We identified 11 drug candidates and analyzed 49,555 MASLD patients (mean age: 63.0 years, SD: 8.6). Atenolol (SHR: 0.81; 95% CI: 0.72–0.92; P<0.001), isosorbide dinitrate (SHR: 0.82; 95% CI: 0.73–0.93; P=0.001), and valsartan (SHR: 0.52; 95% CI: 0.45–0.60; P<0.001) were associated with a reduced risk of cirrhosis. Conversely, amlodipine-based combinations (SHR: 1.24; 95% CI: 1.11–1.39; P<0.001), torasemide (SHR: 1.39; 95% CI: 1.24–1.56; P<0.001), and valsartan-based combinations (SHR: 1.22; 95% CI: 1.09–1.37; P<0.001) were linked to an increased risk.

which permits unrestricted use, distribution, and reproduction in any medium, provided the original author and source are credited.

**Data availability statement:** Data cannot be shared publicly due to the Personal Information Protection Act. However, data are available from the Korea National Health Insurance Service Institutional Data Access/Ethics Committee for researchers who meet the criteria for accessing confidential data. The data underlying the results presented in the study are available from (Korea National Health Insurance Service, https://nhiss.nhis.or.kr).

**Funding:** This research was supported by the Institute of Information & Communications Technology Planning & Evaluation (IITP) grant, funded by the Korean government (MSIT) (No. 2019-0-00224). It was also supported by the MSIT (Ministry of Science and ICT), Korea, under the ICAN (ICT Challenge and Advanced Network of HRD) support program (Grant No. 2710008830), supervised by IITP. Additionally, this work was funded by the Basic Science Research Program through the National Research Foundation of Korea (NRF), supported by the Ministry of Education (No. RS-2024-00461071), the NRF grant funded by the Korean government (MSIT) (RS-2025-00523629), and the Korea University Grant.

**Competing interests:** The authors have declared that no competing interests exist.

## Conclusions

Our findings suggest that antihypertensive drugs such as atenolol and isosorbide dinitrate may protect MASLD patients from cirrhosis, providing valuable insights for clinical applications and treatment strategies.

## Limitations

This study is limited to drugs registered in the Korean NHIS, potentially excluding other relevant candidates. Additionally, the absence of dietary and genetic data in the NHIS database may introduce residual confounding. Lastly, as the study population consists solely of Korean adults, the findings may not be generalizable to other populations.

## Introduction

The global prevalence of non-alcoholic fatty liver disease (NAFLD) continues to escalate, currently standing at 25%, and serves as a cause of liver cirrhosis and hepatocellular carcinoma [1,2]. In 20–30% of cases, NAFLD progresses to non-alcoholic steatohepatitis (NASH), subsequently leading to liver-related events, as well as extrahepatic diseases [3–7].

International experts have recently released a unified statement introducing a new term for fatty liver disease, known as "steatotic liver disease" (SLD) [8–10]. The commonly used term "NAFLD" essentially describes the disease by ruling out other causes such as alcohol and viral infections. Another earlier suggested term, "metabolic dysfunction-associated fatty liver disease" (MAFLD), encompasses all types of SLD from various causes without specifying each cause individually. According to this new statement, SLD is categorized into metabolic dysfunction-associated SLD (MASLD), MASLD with increased alcohol consumption (MetALD), alcohol-related liver disease (ALD), SLD with a specific cause, and cryptogenic SLD [4–6]. MASLD is characterized by the accumulation of liver fat along with at least one of five cardiometabolic risk factors associated with metabolic syndrome with differential nutrients intake [8–10]. In addition, a cross-sectional study has found that MASLD may also be associated with differential nutrients intake [11].

To prevent the progression from fatty liver to liver cirrhosis, several preliminary studies have been conducted. Clifford et al. investigated the use of FXR agonists to treat NAFLD using tissue-specific FXR knockout mice [12]. This study showed how liver FXR controls genes involved in fat creation and how intestinal FXR reduces fat absorption. However, it primarily utilized mouse models to determine the effects of FXR activation on liver TAG, acknowledging limitations in clinical applications as humans and mice have different bile acid types.

In this study, we intend to validate the operational definition of MASLD in the National Health Insurance Service (NHIS) data for several heart failure drugs, including digoxin, by concurrently applying in silico pharmacogenetics and

pharmacoepidemiological approaches. Through the IPTW technique, we aim to adjust for confounding bias to strengthen the evidence on the effects of these drugs for MASLD patients, a high-risk group for liver cirrhosis [13].

## Materials and methods

### Data acquisition

In our study, we embarked on a journey to explore potential therapeutic avenues for MASLD. This involved conducting a comprehensive analysis of gene expression profiles and subsequent drug matching. To achieve this, we utilized a robust dataset from the Gene Expression Omnibus (GEO) database [14]. Our search keywords, including 'NAFLD,' 'non-alcoholic liver disease,' 'NASH,' and 'non-alcoholic steatohepatitis,' yielded 11 distinct datasets: GSE83452, GSE151158, GSE63067, GSE163211, GSE66676, GSE37031, GSE33814, GSE89632, GSE48452, GSE59045, and GSE49541. Using the GEOquery package, we retrieved the GEO dataset and normalized the expression levels of each gene. A linear model was then fitted to the data, and Bonferroni-adjusted P values and logFC (log fold change) were calculated. By utilizing this process, we were able to compare gene expression between patients manifesting pathological indications of simple steatosis and NASH (case group) and those without such indicators (control group). Our primary focus was identifying DEGs between these two groups to uncover molecular signatures associated with non-alcoholic steatotic liver disease and to find potential therapeutic interventions. This method of data extraction and analysis allowed us to deeply understand the molecular mechanisms underlying the disease's pathology, paving the way for developing novel therapeutic strategies.

To delineate the DEGs critical to the disease's progression, we employed stringent selection criteria: an absolute logFC value exceeding 1 and an adjusted P value of below 0.05 [15]. Furthermore, to ensure substantial relevance in the drug matching process, we set a baseline of at least 100 DEGs. Notably, in the GSE89632 dataset, we identified 415 DEGs, comprising 194 up-regulated genes and 221 down-regulated genes, which met these criteria.

### Drug matching

L1000CDS2 is a web-based search engine that utilizes gene expression profiles from the LINCS L1000 database to identify drug candidates that either mimic or reverse a given gene expression signature [16]. This tool takes a list of differentially expressed genes (DEGs) as input and compares it with the extensive dataset of gene expression changes induced by drug treatments in the LINCS L1000 database to calculate the overlap score. Based on this analysis, L1000CDS2 ranks and presents the top 50 drug candidates with the highest potential to either reverse (reverse mode) or mimic (mimic mode) the given disease-associated gene expression pattern. Research using L1000CDS2 has been validated in multiple studies [17,18].

Subsequently, we leveraged the L1000CDS2 web tool to match the 415 identified DEGs with potential therapeutic drugs. The overarching goal of our study was to validate these drug matches using patient data from the NHIS. In this phase, we generated a list of drugs where the Korean ATC code overlapped, focusing primarily on scrutinizing the pharmacodynamics of these drugs.

Remarkably, among the top 50 drugs listed on the L1000CDS2 web tool [16], a drug identified as BRD-K23478508, ranked 46th with an overlap score of 0.1448, emerged as the sole drug with a matching Korean ATC code. This makes it the unique candidate that intersects with the drugs listed in the NHIS. Known as "Digoxin", this drug is now highlighted as a potential therapeutic candidate for MASLD (https://maayanlab.cloud/L1000CDS2/#/result/64feac52b2a2e30055542cbe).

To identify drugs similar to Digoxin, we searched DrugBank [19] for medications with the same indications. Among 26 such drugs, we extracted 12 that are available through the NHIS Korean ATC code. Ultimately, we selected 8 drugs (amlodipine, atenolol, digoxin, furosemide, isosorbide dinitrate, telmisartan, torasemide, and valsartan) that were prescribed to more than 500 individuals between 2013 and 2014. In this context, combination drugs were labeled as 'drug-based combinations' (amlodipine-based, telmisartan-based, and valsartan-based combinations). The workflow of these

steps is described in S1 Fig. The identified drugs were tested in the NHIS database according to the cumulative days of drug prescriptions.

## Study population

The subjects enrolled in this retrospective national cohort study were derived from the health screening cohort of the NHIS. The NHIS provides mandatory healthcare insurance, encompassing medical services, for all citizens of South Korea [20]. This organization collects personalized information including socio-demographics, anthropometric measurements, health screening records, lifestyle surveys, and treatment details, such as hospital attendance and prescribed medications. Several large cohort studies have defined SLD using the NHIS database, and the validity of the data is described in detail elsewhere [4,10,21]. We accessed data from 01/08/2023 for research purposes (NHIS-2023-2-122).

Of the 59,725 non-drinking individuals with MASLD who underwent health screening examinations between 2013 and 2014, we excluded subjects due to death (n = 1,019), missing information for covariates (n = 337), existing liver diseases (n = 6,475), liver transplantation (n = 15), a history of primary liver cancer (n = 561), and a history of liver cirrhosis (n = 1,763) prior to the follow-up investigation (Fig 1). The presence of MASLD was confirmed when a participant exhibited both hepatic steatosis (fatty liver index ≥ 60) and at least one of the five cardiometabolic risk factors, which include (1) a body mass index (BMI) ≥ 23 kg/m$^2$ or waist circumference ≥ 90 cm for men and ≥ 85 cm for women [22], (2) fasting serum glucose ≥ 100 mg/dL, type 2 diabetes, or treatment for type 2 diabetes, (3) blood pressure ≥ 130/85 mmHg or treatment with antihypertensive drugs, (4) triglycerides ≥ 150 mg/dL or lipid-lowering treatment, and (5) high-density lipoprotein (HDL) cholesterol ≤ 40 mg/dL for men and ≤ 50 mg/dL for women or lipid-lowering treatment [8–10].

## Definition of liver cirrhosis

The primary outcome was liver cirrhosis (International Classification of Diseases 10th Revision [ICD-10]: K74, I859, K703, K717, K746, K766, and I982), including decompensated cirrhosis, hepatic encephalopathy, ascites, variceal bleeding, hepatorenal syndrome, and liver failure (ICD-10: G934, R18, I850, I983, K767, K704, K720, K721, and K729; abdominal paracentesis, endoscopic treatment of esophageal or gastric varices, and prescription records for spironolactone,

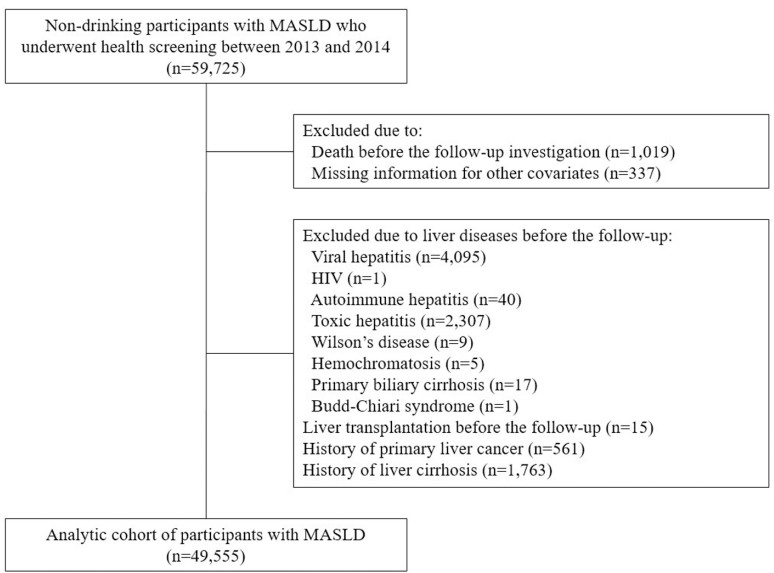

**Fig 1. Flow diagram for the inclusion of the study participants.**

terlipressin, systemic hemostatics, somatostatin, and propranolol). From January 1, 2015, each patient was followed until the earliest instance of liver cirrhosis, death, or December 31, 2019, whichever occurred first.

## Key variables

The following variables were considered as key variables for multivariable analyses: age (continuous; years), sex (categorical; men and women), household income (categorical; upper half and lower half), body mass index (continuous; kg/m$^2$), cigarette smoking (categorical; never, past, and current), moderate-to-vigorous physical activity (categorical; 0, 1–2, 3–4, and ≥5 times/week), a history of cardiovascular disease (categorical; yes and no), and Charlson comorbidity index (CCI; categorical; 0, 1–2, and ≥3). A history of cardiovascular disease was defined using the ICD-10 codes of I20-I25 (coronary heart disease) and I60-I69 (stroke). Cardiovascular disease was excluded from the calculation of the CCI, and the calculation of the CCI was adapted from a previous study [23].

## Statistical analysis

The continuous variables are presented as mean (standard deviation). For the weighted analyses, we used inverse probability of treatment weighting (IPTW) with age, sex, household income, smoking, moderate-to-vigorous physical activity (MVPA), and cardiovascular disease as covariates. The IPTW was conducted separately for each drug, comparing less than 30 cumulative days of prescriptions to 30 or more days. We referred to previous studies that used 30 days or more as the threshold [24]. The between-group standardized mean differences and the distribution of the logit of the propensity scores are shown in S2 Fig and S3 Fig, respectively.

We used the multivariable-adjusted Fine and Gray subdistribution hazard model to calculate the subdistribution hazard ratio (SHR) and estimate the risk of liver cirrhosis with all-cause mortality as a competing event. This was in the context of evaluating the association of specified drugs with the risk of liver cirrhosis, after adjustments for age, sex, household income, BMI, smoking, MVPA, cardiovascular disease, and the Charlson comorbidity index. The multicollinearity of the variables used in the regression model was tested with the variance inflation factor. Proportional hazards assumption was tested using the Schoenfeld residuals. The incidence was calculated by dividing the number of events by 1,000 person-years.

The sensitivity analysis was performed by excluding liver cirrhosis cases that occurred within the first 1, 2, and 3 years from the first date of follow-up to preclude already incident cases. Restricted cubic splines with 4 knots were used to visualize the risk of liver cirrhosis across continuous cumulative days of drug prescriptions. KEGG pathway and network analysis were conducted for Homo sapiens using Entrez ID, with a statistical significance threshold set at $P < 0.05$. DEGs selection and pathway analysis were performed using the R Project for Statistical Computing (version 4.4.2; https://www.r-project.org/). All data collection, mining, and statistical analyses were performed using SAS Enterprise Guide (version 7.3, SAS Institute, Cary, NC, USA).

## Ethics statement

The Institutional Review Board of CHA Bundang Hospital approved this study (No.: CHAMC 2022-04-041). Informed consents were waived because the database was provided for research purposes in an anonymized form under strict confidentiality guidelines.

## Results

### Participant characteristics

The descriptive characteristics of the analytic cohort are shown in Table 1. There were 49,555 MASLD patients with a mean age of 63.0 years (SD, 8.6). The proportions of men and women were 45.3% (n = 22,434) and 54.7%

**Table 1. Baseline characteristics of the study participants who underwent health screening examination between 2013 and 2014.**

| Characteristics | Participants (n = 49,555) |
|---|---|
| Age, years, mean (SD) | 63.04(8.55) |
| Sex, n (%) | |
| Men | 22,434(45.27) |
| Women | 27,121(54.73) |
| Household income, n (%) | |
| Upper half | 30,175(60.89) |
| Lower half | 19,380(39.11) |
| Smoking status, n (%) | |
| Never smoker | 36,755(74.17) |
| Past smoker | 7,876(15.89) |
| Current smoker | 4,924(9.94) |
| MVPA, n (%) | |
| 0 times/week | 26,825(54.13) |
| 1-2 times/week | 6,362(12.84) |
| 3-4 times/week | 6,028(12.16) |
| ≥5 times/week | 10,340(20.87) |
| BMI, kg/m$^2$, mean (SD) | 25.87(2.66) |
| TC, mg/dL, mean (SD) | 201.28(39.93) |
| TG, mg/dL, mean (SD) | 161.1 (88.9) |
| SBP, mm Hg, mean (SD) | 127.28(14.42) |
| FSG, mg/dL, mean (SD) | 105.68(28.21) |
| ALT, IU/L, mean (SD) | 28.17(20.02) |
| Charlson comorbidity index, n (%) | |
| 0 | 18,369(37.07) |
| 1 | 16,063(32.41) |
| ≥2 | 15,123(30.52) |
| Hypertension, n (%) | |
| No | 26,904(54.29) |
| Yes | 22,651(45.71) |
| Type 2 diabetes, n (%) | |
| No | 40,665(82.06) |
| Yes | 8,890(17.94) |
| Dyslipidemia, n (%) | |
| No | 34,791(70.21) |
| Yes | 14,764(29.79) |

Abbreviations: SD, Standard deviation; MVPA, Moderate-to-vigorous physical activity; BMI, Body mass index; TC, Total cholesterol; TG, Triglyceride; SBP, Systolic blood pressure; FSG, Fasting serum glucose; ALT, Alanine aminotransferase.

(n = 27,121), respectively. The mean BMI, total cholesterol, systolic blood pressure, and fasting serum glucose were 25.9 kg/m², 201.3 mg/dL, 127.3 mmHg, and 105.7 mg/dL, respectively. Additionally, the proportions of patients with hypertension, type 2 diabetes, and dyslipidemia were 45.7% (n = 26,904), 17.9% (n = 8,890), and 29.8% (n = 14,764), respectively.

## Performance of the IPTW

For the IPTW, we used age, sex, household income, BMI, smoking, MVPA, and cardiovascular disease as covariates. The standardized mean differences between each drug group and the control group before and after the IPTW are shown in S2 Fig. As illustrated in S3 Fig, the distribution of the logit of the propensity scores was well-fitted between each drug group and the control group after the IPTW.

## Association of drug uses with the risk of liver cirrhosis in patients with MASLD

Crude and minimally adjusted SHRs are shown in S1 Table. In the minimally adjusted model, uses of amlodipine-based combination, digoxin, torasemide, and valsartan-based combination were associated with a higher risk of liver cirrhosis, whereas uses of atenolol, isosorbide dinitrate, and valsartan were associated with a lower risk of liver cirrhosis.

In the fully adjusted model, at least 30 days of using amlodipine-based combination (SHR, 1.24; 95% CI, 1.11–1.39; P<0.001), torasemide (SHR, 1.39; 95% CI, 1.24–1.56; P<0.001), and valsartan-based combination (SHR, 1.22; 95% CI, 1.09–1.37; P<0.001) were associated with a higher risk of liver cirrhosis (Fig 2).

In contrast, at least 30 days of using atenolol (SHR, 0.81; 95% CI, 0.72–0.92; P<0.001), isosorbide dinitrate (SHR, 0.82; 95% CI, 0.73–0.93; P=0.001), and valsartan (SHR, 0.52; 95% CI, 0.45–0.60; P<0.001) were associated with a lower risk of liver cirrhosis. Restricted cubic splines for cumulative days of drug use, which either lowered or raised the risk of liver cirrhosis, include amlodipine (Fig 3A), atenolol (Fig 3B), digoxin (Fig 3C), isosorbide dinitrate (Fig 3D), telmisartan (Fig 3E), and torasemide (Fig 3F).

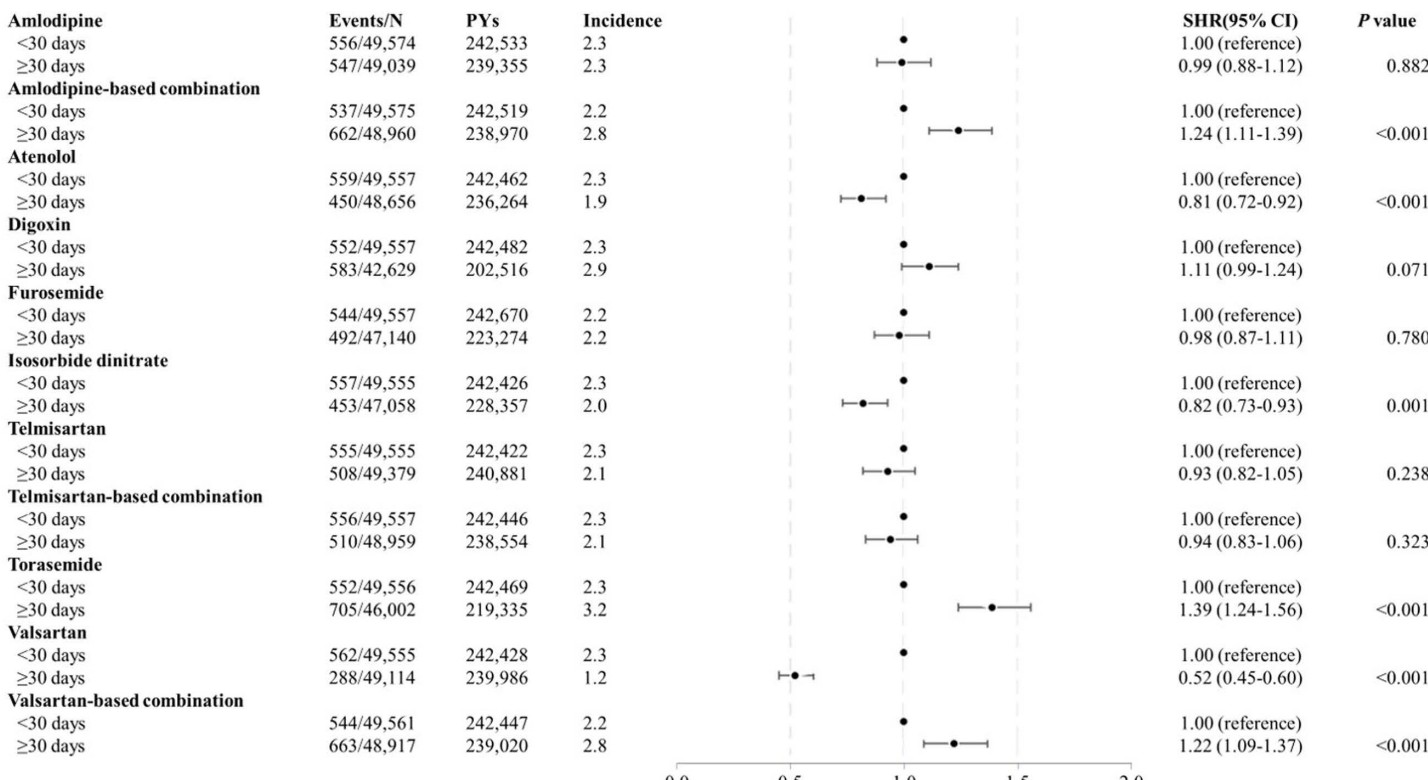

**Fig 2. Forest plot on the risk of liver cirrhosis according to the cumulative days of drug uses.**

**Fig 3. Restricted cubic splines on the risk of liver cirrhosis according to the cumulative days of drug uses.** (A) Amlodipine use. (B) Atenolol use. (C) Digoxin use. (D) Isosorbide dinitrate use. (E) Telmisartan use. (F) Torasemide use.

All these drugs were associated with a lower risk of liver cirrhosis, except for digoxin. While furosemide and valsartan were not significantly associated with the risk of liver cirrhosis (S4 Fig), combination drugs including amlodipine-based, telmisartan-based, and valsartan-based combinations were associated with a lower risk of liver cirrhosis (S5 Fig).

## Sensitivity analysis

Sensitivity analyses were conducted by excluding liver cirrhosis cases that occurred within 1, 2, and 3 years to preclude cases already developing liver cirrhosis, taking into consideration the chronic nature of the disease. All previously identified significant drugs, including amlodipine-based combination, torasemide, valsartan-based combination, atenolol, isosorbide dinitrate, and valsartan, maintained their trends as compared to the results shown in Fig 2 (S2 Table).

## Stratified analysis

Stratified analyses were conducted after dividing participants according to age, sex, obesity, and comorbidities (S3 Table). A significant interaction was identified between age and drug use for digoxin, amlodipine, amlodipine-based combination, valsartan, valsartan-based combination, furosemide, isosorbide dinitrate, and torasemide. Amlodipine, amlodipine-based combination, valsartan, valsartan-based combination, telmisartan-based combination, furosemide, isosorbide dinitrate, and torasemide showed significant interactions by sex in relation to the risk of liver cirrhosis. Additionally, most drugs demonstrated significant interactions with the CCI.

## Pathway and network

The 415 DEGs showed high significance in pathways such as cytokine-cytokine receptor interaction and the TNF-signaling pathway, highlighting the crucial role of immune response and inflammation-related pathways (S6 Fig). These key pathways were connected to numerous genes (EntrezID) and serve as central nodes, demonstrating a high degree of interconnectivity (S7 Fig).

## Discussion

Numerous studies have been conducted to find a new treatment for NAFLD. Holmer et al. [25] proposed primary treatment methods for NAFLD, conducting an RCT on 74 patients. The limitation of this study was the short follow-up period (12 weeks), presenting challenges in predicting the long-term risk of MASLD recurrence post-intervention, particularly due to the difficulty in maintaining short-term weight loss over time.

Takahashi et al. [26] introduced insulin sensitizers (thiazolidinediones) and antioxidants (vitamin E) as the most promising treatments for NAFLD/NASH. However, they noted limitations due to the small number of participants in the RCT, and the use of only one preparation and dosage of probiotic compounds. The study also highlighted the need for a consensus on the most effective and appropriate drug treatment and an understanding the safety and efficacy of long-term medication use.

In our study, we pioneered an innovative approach by leveraging the GEO database, utilizing a signature-based methodology to identify the drug digoxin, and subsequently evaluated the impact of analogous drugs on the incidence risk of liver cirrhosis through a pharmacoepidemiological approach. Our assessment revealed that, even after mitigating potential confounding biases using IPTW, drugs such as atenolol, isosorbide dinitrate, and valsartan demonstrated a protective effect against liver cirrhosis. Atenolol is a selective beta-1 adrenergic receptor blocker that reduces heart rate and cardiac output, thereby lowering blood pressure [27]. This hemodynamic stability is crucial in patients with liver diseases [28,29]. In cirrhotic patients, increased portal pressure can lead to complications [30]; beta-blockers like atenolol help manage portal hypertension by controlling heart rate.

Similarly, isosorbide dinitrate is an organic nitrate that releases nitric oxide (NO) in vivo, promoting vasodilation through smooth muscle relaxation [31,32]. This vasodilatory effect contributes to a reduction in portal pressure, which is particularly relevant in patients with cirrhosis [33]. Portal hypertension is a major driver of complications in chronic liver disease and is closely linked to fibrosis progression. In the context of NAFLD/NASH, targeting intrahepatic vascular resistance may offer therapeutic benefits by modulating portal hemodynamics [30].

Conversely, amlodipine-based combinations, torasemide, and valsartan-based combinations were associated with an elevated risk. To ensure applicability in real-world clinical settings, competing risks were accounted for by considering the possibility that alternative events, such as death, could preclude the occurrence of liver cirrhosis. This approach allows for the assessment of cumulative incidence rather than cause-specific event risk, making it more useful for realistic prognosis evaluation. In scenarios where patients with MASLD present with cardiovascular conditions, among others, our study can provide scientific evidence on which drugs should be utilized, considering the risk aspects of liver cirrhosis. This offers insights that can guide therapeutic strategies and decision-making in clinical settings.

According to Nababan et al. [34], portal hypertension in NAFLD mostly occurs in the cirrhotic stage and is the main factor causing complications of liver cirrhosis. Additionally, Fonseca et al. [35], suggest that improvements in glucose and lipid metabolism mediated by vasodilating beta-blockers like isosorbide dinitrate may help reduce the risk of coronary artery disease in high-risk patients with hypertension. Based on these conventional studies, we can infer that the drugs indicated for hypertension, such as atenolol and isosorbide dinitrate, may be associated with beneficial lipid metabolism against the development of liver cirrhosis.

The first limitation is that, to match with the drugs registered in the Korean NHIS, we selected digoxin as it was the top-ranked drug (overlap score=0.1448). However, this finding could be coincidental. The second limitation is that dietary and genetic information is not available in the Korea NHIS database, which may result in confounding bias due to unmeasured

potential confounders. The third limitation is that this study was conducted on a Korean adult population, which limits its generalizability. Therefore, further studies involving diverse ethnic and racial groups are needed. Lastly, the study participants were individuals who underwent health examinations, which may introduce selection bias, as they might have greater health awareness or better healthcare access compared to those who did not undergo health examinations.

The new criteria employed in this study appear to be more stringent, requiring at least one cardiometabolic risk factor for the classification of metabolic SLD, as compared to the previous definitions which required at least two such factors for defining metabolic dysfunction in MAFLD [8,36]. By not including HOMA-IR and C-reactive protein in defining metabolic dysfunction, these criteria aim to be more universally adaptable and conducive to broader assessments of SLD status, given the relative standardization and frequency of glycemic and lipid measurements. Under these refined parameters, MASLD becomes significant for the insights it provides into the interlinked roles of metabolic dysfunctions and hepatic conditions in cardiovascular risk profiles, offering a refined perspective to understand the interconnected pathways leading to cardiovascular incidents.

In summary, our study has devised a high-throughput method for pinpointing potential drug repurposing candidates by leveraging gene expression patterns, with subsequent validation of these candidates using clinical NHIS data. We hope that our study provides treatment options for MASLD patients that warrants further assessment through additional animal safety assessments and forthcoming clinical trials.

## Supporting information

**S1 Fig. Workflow before validation using NHIS cohort data.**
(TIF)

**S2 Fig. Standardized mean differences between the treated and control groups before and after the IPTW according to the specified drug use.** Treated group indicates at least 30 days of cumulative days of the specified drug use. (A) Amlodipine. (B) Amlodipine-based combination. (C) Atenolol. (D) Digoxin. (E) Furosemide. (F) Isosorbide dinitrate. (G) Telmisartan. (H) Telmisartan-based combination. (I) Torasemide. (J) Valsartan. (K) Valsartan-based combination.
(TIF)

**S3 Fig. Logit of propensity scores of the treated and control groups before and after the IPTW according to the specified drug use.** (A) Amlodipine. (B) Amlodipine-based combination. (C) Atenolol. (D) Digoxin. (E) Furosemide. (F) Isosorbide dinitrate. (G) Telmisartan. (H) Telmisartan-based combination. (I) Torasemide. (J) Valsartan. (K) Valsartan-based combination.
(TIF)

**S4 Fig. Restricted cubic splines on the risk of liver cirrhosis according to the cumulative days of furosemide and valsartan uses.** (A) Furosemide. (B) Valsartan.
(TIF)

**S5 Fig. Restricted cubic splines on the risk of liver cirrhosis according to the cumulative days of combination drugs.** (A) Amlodipine-based combination. (B) Telmisartan-based combination. (C) Valsartan-based combination.
(TIF)

**S6 Fig. KEGG pathway enrichment analysis based on the input DEGs.** Dot plot showing the enriched KEGG pathways based on the input gene set. The y-axis lists the names of the significantly enriched pathways, while the x-axis represents the gene ratio. Dot size corresponds to the number of genes mapped to each pathway, and dot color indicates the adjusted p-value, with red representing more significant enrichment.
(TIF)

**S7 Fig. Gene network of significantly enriched KEGG pathway.** This network plot visualizes the relationships between significantly enriched KEGG pathways and their associated genes. Orange nodes represent KEGG pathways, and gray nodes represent individual genes. Edges indicate that a gene is involved in the connected pathway. The size of each pathway node reflects the number of genes it contains.
(TIF)

**S1 Table. Minimally adjusted subdistribution hazard ratios for drug use against the risk of liver cirrhosis.**
(DOCX)

**S2 Table. Sensitivity analysis on the risk of liver cirrhosis according to the cumulative days of drug use after excluding events that occurred within specified periods.**
(DOCX)

**S3 Table. Stratified analysis on the risk of liver cirrhosis according to the cumulative days of drug use.**
(DOCX)

## Author contributions

**Conceptualization:** Seogsong Jeong.

**Data curation:** Chae Won Lee, Eun Seok Kang.

**Formal analysis:** Eun Seok Kang, Seogsong Jeong.

**Funding acquisition:** Hyun Wook Han.

**Investigation:** Chae Won Lee, Eun Seok Kang.

**Methodology:** Chae Won Lee, Seogsong Jeong.

**Software:** Eun Seok Kang, Seogsong Jeong.

**Validation:** Eun Seok Kang, Seogsong Jeong.

**Visualization:** Eun Seok Kang, Seogsong Jeong.

**Writing – original draft:** Chae Won Lee, Eun Seok Kang, Seogsong Jeong.

**Writing – review & editing:** Seogsong Jeong, Hyun Wook Han.

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
