## [Decision Letter · Decision Letter 0]

11 Mar 2025

PONE-D-24-48206In Silico Drug Repositioning-based Identification of Potentially Effective Drugs for Metabolic Dysfunction-Associated Steatotic Liver Disease against Liver CirrhosisPLOS ONE

Dear Dr. Han,

Thank you for submitting your manuscript to PLOS ONE. After careful consideration, we feel that it has merit but does not fully meet PLOS ONE’s publication criteria as it currently stands. Therefore, we invite you to submit a revised version of the manuscript that addresses the points raised during the review process.

We look forward to receiving your revised manuscript.

Kind regards,

Samuel O. Antwi, Ph.D.

Academic Editor

PLOS ONE

Journal Requirements:

Reviewers' comments:

Reviewer's Responses to Questions

**Comments to the Author**

1. Is the manuscript technically sound, and do the data support the conclusions?

Reviewer #1: Yes

Reviewer #2: Yes

Reviewer #3: Partly

2. Has the statistical analysis been performed appropriately and rigorously? 

Reviewer #1: Yes

Reviewer #2: Yes

Reviewer #3: Yes

3. Have the authors made all data underlying the findings in their manuscript fully available?

Reviewer #1: Yes

Reviewer #2: Yes

Reviewer #3: No

4. Is the manuscript presented in an intelligible fashion and written in standard English?

Reviewer #1: Yes

Reviewer #2: Yes

Reviewer #3: Yes

5. Review Comments to the Author

Reviewer #1: Title & Abstract:

The abstract should explicitly state study limitations (e.g., observational design constraints) and provide more details on the in silico methodology.

Methods:

While the methods are rigorous, the manuscript should provide more details on handling multicollinearity and proportional hazard assumptions in the statistical models.

The description of the Korean NHIS database should include specifics about missing data handling and validation.

Results:

The results are clearly presented, but the clinical significance of SHR values should be discussed further.

Include subgroup-specific results in tables or figures to improve transparency.

Discussion:

The discussion highlights the significance of the findings but should expand on the biological mechanisms linking the identified drugs to reduced cirrhosis risk.

Limitations such as residual confounding and generalizability should be explicitly acknowledged.

Data Availability:

Provide clearer guidance on how external researchers can access the data through the NHIS committee.

Reviewer #2: The manuscript addresses a question of significant clinical and epidemiological relevance in a robust and methodological manner, focusing on the association of certain potential medications for the treatment of metabolic dysfunction-associated steatotic liver disease (MASLD), previously known as non-alcoholic fatty liver disease (NAFLD), and its progression to liver cirrhosis.

However, some improvements could be incorporated into the manuscript to enhance its solidity, as outlined below:

The manuscript employs an innovative and robust methodological approach. The adjustment for confounding factors and the application of advanced statistical techniques (IPTW, Fine and Gray models) strengthen the validity of the results. Nevertheless, certain aspects could be improved, such as:

-The choice of a logFC >1 and p<0.05 as criteria for gene selection could be better justified. For instance, why was a score of 0.1448 deemed sufficient to prioritize digoxin?

-Although the article mentions the identification of differentially expressed genes (DEGs) from GEO datasets, I believe some details are lacking regarding how these genes were prioritized.

-It should be explicitly stated why the cutoff point was chosen based solely on prescriptions exceeding 30 days.

-The discussion on the mechanisms by which the identified drugs (atenolol, isosorbide dinitrate) may protect against liver cirrhosis is somewhat limited. A more in-depth review of the literature on the metabolic and hepatic effects of these drugs would enrich the discussion.

The article presents a solid methodology and clinically relevant results but could benefit from a more thorough discussion of the limitations and underlying mechanisms of the findings. By implementing these improvements, the article will have a high impact on the global scientific and academic community.

Reviewer #3: Based on review of clinical health screening data, Authors propose a signature-based in silico approach to identify specific drugs that could be used to mitigate a risk of liver cirrhosis in MASLD patients. To the best of my knowledge, the study presents original research that have not been published elsewhere. Despite the evidence provided, the manuscript has several areas that require addressing.

It is unclear how the identified 415 differentially expressed genes impacted the identification of digoxin – can the authors please provide more specific information and some additional clarity on that finding? No information is provided on the DEGs that were up/down-regulated, the pathways they impact, or any commonality/relatedness between them.

The initial identification of digoxin appears somewhat random and fortuitous, as it was the only drug with a matching Korean ATC code. The basis of additional related-drug selections followed on from that initial ‘identification’. Can the authors please provide some clarity on whether any hypotheses were generated before assessing these databases on the potential drugs and/or their related mechanism of action that they expected may be implicated?

Please provide specifics on how “a meticulous validation study to assess the prospective efficacy of Digoxin using extensive patient data from the NHIS” will be performed? Also, the data provided herein indicated that use of Digoxin is associated with a higher risk of liver cirrhosis, not lower, therefore the suggestion to evaluate its ‘prospective efficacy’ seems unwarranted.

In essence, this paper is hypothesis generating, based on currently available database information, without any formal validation performed, which the authors contend that this will be provided in a follow up study. While the association data provided on the potential impact of atenolol, isosorbide dinitrate and valsartan on lowered risk of liver cirrhosis are of interest, it is unclear based on the information provided in the manuscript how the signature-based methodology played any part in supporting these findings. Additional clarity on the exact methods used to support this are crucial to supporting the overall potential of such in silico methodologies in identifying potentially effective drugs to MASLD.

6. PLOS authors have the option to publish the peer review history of their article (what does this mean? ). If published, this will include your full peer review and any attached files.

**Do you want your identity to be public for this peer review?** For information about this choice, including consent withdrawal, please see our Privacy Policy .

Reviewer #1: **Yes: ** Silvia Sovaila

Reviewer #2: No

Reviewer #3: No

---

## [Author Response · Author response to Decision Letter 1]

27 Mar 2025

March 27, 2025

RE: PLOS ONE

Attn: Samuel O. Antwi, Ph.D. Academic Editor

Title: Revised manuscript Submission ID PONE-D-24-48206 submission to PLOS ONE

Dear Professor Samuel,

Your letter dated March 11, 2025, is greatly appreciated. We are grateful for the opportunity to submit this revised manuscript to PLOS ONE. We have provided a point-by-point response to the academic editor's and reviewers' comments. Major changes that were made for the revision are shown in red color. Additionally, we have added new supplementary figures 6 and 7.

We are pleased to resubmit our revised manuscript for possible publication in PLOS ONE. Your favorable consideration would be greatly appreciated.

Sincerely Yours.

Seogsong Jeong and Hyun Wook Han

[Response to journal requirements]

Comment 1: Please ensure that your manuscript meets PLOS ONE's style requirements, including those for file naming. The PLOS ONE style templates can be found at

Author’s response: Thank you for your guidance. According to your instructions, we have adjusted our manuscript to fit PLOS ONE’s style.

Comment 2: We note that you have indicated that there are restrictions to data sharing for this study. PLOS only allows data to be available upon request if there are legal or ethical restrictions on sharing data publicly. For more information on unacceptable data access restrictions, please see http://journals.plos.org/plosone/s/data-availability#loc-unacceptable-data-access-restrictions.

Author’s response: Thank you for your comments. We have updated our Data Availability statement in the manuscript following PLOS ONE’s criteria. The revised details are as follows.

[Page 18, ‘Data availability’ section]

“Data cannot be shared publicly due to the personal information protection act. However, data are available from the Korea National Health Insurance Service Institutional Data Access/Ethics Committee for researchers who meet the criteria for accessing confidential data. The data underlying the results presented in the study are available from (Korea National Health Insurance Service, https://nhiss.nhis.or.kr).”

Comment 3: Please include captions for your Supporting Information files at the end of your manuscript, and update any in-text citations to match accordingly. Please see our Supporting Information guidelines for more information: http://journals.plos.org/plosone/s/supporting-information.

Author’s response: Thank you for your guidance. Following your instructions, we have reviewed our captions for the supporting information files and corrected them to fit PLOS ONE’s guidelines.

[Response to Reviewer’s comments]

Reviewer #1

Comment 1: Title & Abstract: The abstract should explicitly state study limitations (e.g., observational design constraints) and provide more details on the in silico methodology.

Author’s response: Thank you for your thoughtful comment. In response to your suggestion, we have revised the abstract to include the study's limitations and the in silico methodology as follows.

[Page 2, Line 36-39, ‘Limitations’ section in Abstract]

“This study is limited to drugs registered in the Korean NHIS, potentially excluding other relevant candidates. Additionally, the absence of dietary and genetic data in the NHIS database may introduce residual confounding. Lastly, as the study population consists solely of Korean adults, the findings may not be generalizable to other populations.”

[Page 2, Line 21-26, ‘Methods’ section in Abstract]

“We analyzed gene expression datasets to identify differentially expressed genes (DEGs) in MASLD and matched them to candidate drugs using L1000CDS2. We further validated potential drugs by cross-referencing with prescription data from the Korea National Health Insurance Service (NHIS). Participants who underwent health screenings between 2013 and 2014 were included. MASLD was diagnosed in individuals with hepatic steatosis (fatty liver index ≥60) and at least one cardiometabolic risk factor.”

Comment 2: Methods: While the methods are rigorous, the manuscript should provide more details on handling multicollinearity and proportional hazard assumptions in the statistical models.

The description of the Korean NHIS database should include specifics about missing data handling and validation.

Author’s response: Thank you for your valuable feedback. We have added the following detailed explanation on handling multicollinearity and the proportional hazards assumption.

[Page 10, Line 174-176]

“The multicollinearity of the variables used in the regression model was tested with the variance inflation factor. Proportional hazards assumption was tested using the Schoenfeld residuals.”

Additionally, we have specified the following details on missing data handling and validation in the description of the Korea NHIS database.

[Page 8, Line 129-133]

“Of the 59,725 non-drinking individuals with MASLD who underwent health screening examinations between 2013 and 2014, we excluded subjects due to death (n=1,019), missing information for covariates (n=337), existing liver diseases (n=6,475), liver transplantation (n=15), a history of primary liver cancer (n=561), and a history of liver cirrhosis (n=1,763) prior to the follow-up investigation (Fig 1).”

[Page 9, Line 170-174]

“We used the multivariable-adjusted Fine and Gray subdistribution hazard model to calculate the subdistribution hazard ratio (SHR) and estimate the risk of liver cirrhosis with all-cause mortality as a competing event. This was in the context of evaluating the association of specified drugs with the risk of liver cirrhosis, after adjustments for age, sex, household income, BMI, smoking, MVPA, cardiovascular disease, and the Charlson comorbidity index.”

Comment 3: Results: The results are clearly presented, but the clinical significance of SHR values should be discussed further.

Include subgroup-specific results in tables or figures to improve transparency.

Author’s response: Thank you for your valuable comments. We have added the clinical significance of the SHR values to the Method section, and the subgroup analysis has now been presented in S3 Table. The revised details are as follows.

[Page 16, Line 293-296]

“To ensure applicability in real-world clinical settings, competing risks were accounted for by considering the possibility that alternative events, such as death, could preclude the occurrence of liver cirrhosis. This approach allows for the assessment of cumulative incidence rather than cause-specific event risk, making it more useful for realistic prognosis evaluation.”

Comment 4: Discussion: The discussion highlights the significance of the findings but should expand on the biological mechanisms linking the identified drugs to reduced cirrhosis risk.

Limitations such as residual confounding and generalizability should be explicitly acknowledged.

Author’s response: Thank you for your valuable comments. We have added the biological mechanism of the identified drugs in the discussion section as follows.

[Page 15, Line 282-291]

“Atenolol is a selective beta-1 adrenergic receptor blocker that reduces heart rate and cardiac output, thereby lowering blood pressure [27]. This hemodynamic stability is crucial in patients with liver diseases [28, 29]. In cirrhotic patients, increased portal pressure can lead to complications [30]; beta-blockers like atenolol help manage portal hypertension by controlling heart rate.

Similarly, isosorbide dinitrate is an organic nitrate that releases nitric oxide (NO) in vivo, promoting vasodilation through smooth muscle relaxation [31, 32]. This vasodilatory effect contributes to a reduction in portal pressure, which is particularly relevant in patients with cirrhosis [33]. Portal hypertension is a major driver of complications in chronic liver disease and is closely linked to fibrosis progression. In the context of NAFLD/NASH, targeting intrahepatic vascular resistance may offer therapeutic benefits by modulating portal hemodynamics [30].”

Furthermore, we have added the following points regarding the bias related to potential confounding factors and the limitations in generalizability.

[Page 16, Line 310-316]

“The second limitation is that dietary and genetic information is not available in the Korea NHIS database, which may result in confounding bias due to unmeasured potential confounders. The third limitation is that this study was conducted on a Korean adult population, which limits its generalizability. Therefore, further studies involving diverse ethnic and racial groups are needed. Lastly, the study participants were individuals who underwent health examinations, which may introduce selection bias, as they might have greater health awareness or better healthcare access compared to those who did not undergo health examinations.”

Comment 5: Data Availability: Provide clearer guidance on how external researchers can access the data through the NHIS committee.

Author’s response: Thank you for your comments. We have updated our Data Availability statement in the manuscript following PLOS ONE’s criteria. The revised details are as follows.

[Page 18, ‘Data availability’ section]

“Data cannot be shared publicly due to the personal information protection act. However, data are available from the Korea National Health Insurance Service Institutional Data Access/Ethics Committee for researchers who meet the criteria for accessing confidential data. The data underlying the results presented in the study are available from (Korea National Health Insurance Service, https://nhiss.nhis.or.kr).”

Reviewer #2

The manuscript addresses a question of significant clinical and epidemiological relevance in a robust and methodological manner, focusing on the association of certain potential medications for the treatment of metabolic dysfunction-associated steatotic liver disease (MASLD), previously known as non-alcoholic fatty liver disease (NAFLD), and its progression to liver cirrhosis.

However, some improvements could be incorporated into the manuscript to enhance its solidity, as outlined below:

The manuscript employs an innovative and robust methodological approach. The adjustment for confounding factors and the application of advanced statistical techniques (IPTW, Fine and Gray models) strengthen the validity of the results. Nevertheless, certain aspects could be improved, such as:

Comment 1: The choice of a logFC >1 and p<0.05 as criteria for gene selection could be better justified. For instance, why was a score of 0.1448 deemed sufficient to prioritize digoxin?

Author’s response: Thank you for your constructive comments. A logFC, which represents the logarithm (typically base 2) of the expression ratio between two conditions, is considered significant when its absolute value exceeds 1 (indicating at least a twofold change in expression), while a p-value below 0.05 signifies statistical significance by indicating that the observed difference is unlikely to have occurred by chance. In various studies, a logFC absolute value of >1 is widely used as a threshold for up-regulation and down-regulation, and a p value of less than 0.05 is often applied as a significance criterion [Meiners, Franziska et al. “Computational identification of natural senotherapeutic compounds that mimic dasatinib based on gene expression data.” Scientific reports vol. 14,1 6286. 15 Mar. 2024, doi:10.1038/s41598-024-55870-4.] [Papadopoulou, Dimitra et al. “Repurposing the antipsychotic drug amisulpride for targeting synovial fibroblast activation in arthritis.” JCI insight vol. 8,9 e165024. 8 May. 2023, doi:10.1172/jci.insight.165024]. The baseline content referenced ‘Reference 15’.

The overlap score is calculated by dividing the number of intersecting genes by the total number of input genes, assessing the extent of which a specific drug suppresses genes that are overexpressed in a disease state that increases genes that are underexpressed. A higher score indicates a stronger association between the drug and the given gene signature.

For validation, we examined drug was digoxin, with an overlap score of 0.1448. This overlap score has been studied in previous study at similar or even lower values. [Meiners, Franziska et al. “Computational identification of natural senotherapeutic compounds that mimic dasatinib based on gene expression data.” Scientific reports vol. 14,1 6286. 15 Mar. 2024, doi:10.1038/s41598-024-55870-4.] [Papadopoulou, Dimitra et al. “Repurposing the antipsychotic drug amisulpride for targeting synovial fibroblast activation in arthritis.” JCI insight vol. 8,9 e165024. 8 May. 2023, doi:10.1172/jci.insight.165024].

However, since the discovery of digoxin could be incidental, we included the following limitations. Moreover, we have added the URL containing the matching process from L1000CDS2 to the main text, and this URL includes information on up-regulated and down-regulated genes.

[Page 16, Line 308-310]

“The first limitation is that, to match with the drugs registered in the Korean NHIS, we selected digoxin as it was the top-ranked drug (overlap score=0.1448). However, this finding could be coincidental.”

[Page 7, Line 110-111]

“Known as “Digoxin”, this drug is now highlighted as a potential therapeutic candidate for MASLD (https://maayanlab.cloud/L1000CDS2/#/result/64feac52b2a2e30055542cbe).”

Comment 2: Although the article mentions the identification of differentially expressed genes (DEGs) from GEO datasets, I believe some details are lacking regarding how these genes were prioritized.

Author’s response: Thank you for your constructive comments. We have strengthened the explanation of the selection process for the 415 genes as follows.

[Page 6, Line 79-83]

“Using the GEOquery package, we retrieved the GEO dataset and normalized the expression levels of each gene. A linear model was then fitted to the data, and Bonferroni-adjusted P values and logFC (log fold change) were calculated. By utilizing this process, we were able to compare gene expression between patients manifesting pathological indications of simple steatosis and NASH (case group) and those without such indicators (control group).”

Comment 3: It should be explicitly stated why the cutoff point was chosen based solely on prescriptions exceeding 30 days.

Author’s response: There have been studies analyzing digoxin by dividing it into short-term (30 days) and long-term (over one year) use [Qamer, Syed Z et al. “Digoxin Use and Outcomes in Patients With Heart Failure With Reduced Ejection Fraction.” The American journal of medicine vol. 132,11 (2019): 1311-1319. doi:10.1016/j.amjmed.2019.05.012].

[Page 9, Line 167]

“We referred to previous studies that used 30 days or more as the threshold [24].”

Comment 4: The discussion on the mechanisms by which the identified drugs (atenolol, isosorbide dinitrate) may protect against live

---

## [Decision Letter · Decision Letter 1]

16 Apr 2025

Identification of potentially effective drugs for metabolic dysfunction-associated steatotic liver disease against liver cirrhosis: In-silico drug repositioning-based retrospective cohort study

PONE-D-24-48206R1

Dear Dr. Han,

We’re pleased to inform you that your manuscript has been judged scientifically suitable for publication and will be formally accepted for publication once it meets all outstanding technical requirements.

Kind regards,

Samuel O. Antwi, Ph.D.

Academic Editor

PLOS ONE

Additional Editor Comments (optional):

Reviewers' comments:

Reviewer's Responses to Questions

**Comments to the Author**

1. If the authors have adequately addressed your comments raised in a previous round of review and you feel that this manuscript is now acceptable for publication, you may indicate that here to bypass the “Comments to the Author” section, enter your conflict of interest statement in the “Confidential to Editor” section, and submit your "Accept" recommendation.

Reviewer #1: All comments have been addressed

Reviewer #3: All comments have been addressed

2. Is the manuscript technically sound, and do the data support the conclusions?

Reviewer #1: Yes

Reviewer #3: Yes

3. Has the statistical analysis been performed appropriately and rigorously? 

Reviewer #1: Yes

Reviewer #3: N/A

4. Have the authors made all data underlying the findings in their manuscript fully available?

Reviewer #1: Yes

Reviewer #3: Yes

5. Is the manuscript presented in an intelligible fashion and written in standard English?

Reviewer #1: Yes

Reviewer #3: Yes

6. Review Comments to the Author

Reviewer #1: The manuscript is suitable for acceptance after the revisions, effectively addressing the previously raised concerns and significantly strengthening the overall robustness of the findings.

Reviewer #3: I acknowledge that the authors have addressed my comments and provided additional language, references and clarity on the methods used to support the manuscripts conclusions, while also acknowledging some of the limitations in the updated manuscript. I thank the authors for their responses.

7. PLOS authors have the option to publish the peer review history of their article (what does this mean? ). If published, this will include your full peer review and any attached files.

**Do you want your identity to be public for this peer review?** For information about this choice, including consent withdrawal, please see our Privacy Policy .

Reviewer #1: **Yes: ** Silvia Sovaila

Reviewer #3: No

---

## [Editor Report · Acceptance letter]

PONE-D-24-48206R1

PLOS ONE

Dear Dr. Han,

I'm pleased to inform you that your manuscript has been deemed suitable for publication in PLOS ONE. Congratulations! Your manuscript is now being handed over to our production team.

Kind regards,

on behalf of

Dr. Samuel O. Antwi

Academic Editor

PLOS ONE